# Selective Delivery of Tofacitinib Citrate to Hair Follicles Using Lipid-Coated Calcium Carbonate Nanocarrier Controls Chemotherapy-Induced Alopecia Areata

**DOI:** 10.3390/ijms24098427

**Published:** 2023-05-08

**Authors:** Yeneng Guan, Aqin Yan, Wei Qiang, Rui Ruan, Chaobo Yang, Kai Ma, Hongmei Sun, Mingxing Liu, Hongda Zhu

**Affiliations:** Cooperative Innovation Center of Industrial Fermentation (Ministry of Education & Hubei Province), Key Laboratory of Fermentation Engineering (Ministry of Education), National “111” Center for Cellular Regulation and Molecular Pharmaceutics, School of Food and Biological Engineering, Hubei University of Technology, Wuhan 430068, China

**Keywords:** chemotherapy-induced alopecia (CIA), tofacitinib citrate (TFC), phospholipid-calcium carbonate hybrid nanoparticles, follicular-targeted

## Abstract

Chemotherapy-induced alopecia (CIA) is one of the common side effects in cancer treatment. The psychological distress caused by hair loss may cause patients to discontinue chemotherapy, affecting the efficacy of the treatment. The JAK inhibitor, Tofacitinib citrate (TFC), showed huge potential in therapeutic applications for treating baldness, but the systemic adverse effects of oral administration and low absorption rate at the target site limited its widespread application in alopecia. To overcome these problems, we designed phospholipid-calcium carbonate hybrid nanoparticles (PL/ACC NPs) for a topical application to target deliver TFC. The results proved that PL/ACC-TFC NPs showed excellent pH sensitivity and transdermal penetration in vitro. PL/ACC NPs offered an efficient follicular targeting approach to deliver TFC in a Cyclophosphamide (CYP)-induced alopecia areata mouse model. Compared to the topical application of TFC solution, PL/ACC-TFC NPs significantly inhibited apoptosis of mouse hair follicles and accelerated hair growth. These findings support that PL/ACC-TFC NPs has the potential for topical application in preventing and mitigating CYP-induced Alopecia areata.

## 1. Introduction

Alopecia areata (AA) is a sudden onset non-scar inflammatory hair loss disease, with the second highest incidence rate in the hair loss category. Although it is not a life-threatening disease, it can affect the appearance of a patient and cause serious disturbances to the patient’s life [1]. Influencing factors of AA include genetics, environmental factors, and drug stimulation, which may lead to the immune system attacking hair follicles (HF) and causing reversible hair loss [2]. Chemotherapy-induced alopecia (CIA) is one of the most common side effects in cancer treatment, as is hair loss caused by drug stimulation [3]. Some patients, especially women, may refuse or discontinue chemotherapy because of psychological distress induced by hair loss. It has been reported that over 50% of breast cancer patients reckoned that hair loss was the heaviest psychological burden during perioperative chemotherapy [4]. Thus, establishing methods to prevent and alleviate CIA could help reduce anxiety in patients and improve the efficacy of chemotherapy.

As for the common treatments of AA, sebaceous steroids, photochemical therapy, and immunomodulators have some disadvantages, including undesired adverse effects and high recurrence rates [5]. Physical therapy, e.g., scalp hypothermia, has been reported for the treatment of CIA, but metastasis of the scalp is a potential risk. Therefore, applying it to cancer patients was not recommended [6]. It was evident that Janus kinase (JAK) inhibitors, such as Tofacitinib (TFC), showed considerable potential in therapeutic applications for treating baldness [7]. However, oral administration of Tofacitinib might induce systemic adverse effects, such as infection, anemia, nausea and vomiting, and even risk of blood clots [8]. When TFC is used for non-life-threatening diseases such as hair loss, it should involve topical administration to avoid the adverse effects of systemic administration. However, it should be noted that the barrier effect of the skin stratum corneum limits active molecules to reach the target site. The topical application of the nanoparticle drug delivery system can enhance the penetration of hair follicles, and the funnel-like structure of the hair follicle acts as a warehouse; therefore, particulate matter can penetrate deeper into the follicle and accumulate more [9]. It has been reported that phospholipid-polymer hybrid nanoparticles exhibited excellent delivery capabilities as targeted delivery carriers for hair follicles (HFs) [10,11]. TFC has been designed and loaded into different nanocarriers, such as polymer micelles [10] and nanoparticles [12], to enhance the therapeutic effects of TFC for AA.

In this study, we designed phospholipid-calcium carbonate hybrid nanoparticles to deliver TFC (PL/ACC-TFC NPs). This nanocarrier might penetrate through the stratum corneum of the skin and target TFC delivery to the hair follicles. As shown in Figure 1, since the template of PL/ACC-TFC was amorphous ACC NPs with pH sensitivity, it could control release within the weak acidic pH of the hair follicle [13]. In addition, the CO_2_ generated by the decomposition of ACC provided a driving force to enhance the permeability of TFC, improving the penetration depth of TFC in the hair follicle (Figure 1). During the formation of CIA, the inflammatory factor IFN-γ (secreted by CD8+NKG2D+ T cells) increased due to chemotherapy stimulation. IFN-γ could bind to the receptors of hair follicle epithelial cells, resulting in JAK1/2-mediated STAT1 phosphorylation, thereby causing the hair follicle epithelial cell nucleus to secrete more ectopic MHC class I and II molecules. These molecules would recruit more CD8+NKG2D+ T cells to attack hair follicles and lead to hair loss [1]. TFC delivered by PL/ACC NPs could interfere with this feedback loop to relieve symptoms of hair loss. Herein, the properties of PL/ACC-TFC NPs were evaluated to investigate the potential of hair follicle targeting and penetration of PL/ACC NPs in vitro. The effect of PL/ACC-TFC on hair growth in a cyclophosphamide (CYP)-induced alopecia areata mouse model was evaluated, and the results revealed the potential application of PL/ACC-TFC NPs to prevent and alleviate CIA.

## 2. Results and Discussion

### 2.1. Preparation and Characterization of PL/ACC-TFC

Calcium carbonate-based lipid nanoparticles loaded with TFC (PL/ACC-TFC NPs) of different sizes were designed. As shown in Appendix A, amorphous calcium carbonate nanoparticles (denoted as ACC NPs) with different particle sizes were prepared via the gas diffusion method [14], and then TFC was loaded into ACC NPs through physisorption to form ACC-TFC NPs. Because amorphous ACC NP was easy to decompose when exposed to an aqueous environment, phospholipid bilayer was modified around ACC-TFC NPs to fabricate PL/ACC-TFC NPs, which could improve its stability, biocompatibility, and follicle targeting [15]. From ACC NPs to PL/ACC-TFC NPs, the size of nanoparticles increased from 200.6 ± 14.7 nm to 285.1 ± 8.0 nm (DLS analysis), and polydispersity index (PDI) increased from 0.06 ± 0.02 to 0.28 ± 0.04 due to the loaded TFC and coat phospholipid bilayer (Figure 2a). The reversal of Zeta potential of PL/ACC-TFC NPs from 16.8 ± 1.2 mV to −29.8 ± 1.3 mV indicated successful phospholipid coating (Figure 2b). Lipid nanoparticles with a size of 200–400 nm might more efficiently penetrate hair follicles compared to non-particulate substances [16], and our results of penetration evaluation in vitro also confirmed that PL/ACC-TFC NPs with a particle size of about 280 nm had better dermal and follicular permeability than PL/ACC-TFC NPs with a particle size of 150 nm or 460 nm (Figure 2f). Therefore, ACC NPs with a particle size of about 200 nm were chosen to load TFC and coat the phospholipid layer, which was stable during storage at 4 °C for 30 days (Figure 2d) and had the highest steady-state transdermal rate (Figure 2g). The TEM image of a PL/ACC-TFC NP showed that it had a uniform spherical structure with a particle size of 187.3 nm, though its size decreased due to drying upon comparison with DLS analysis (Figure 2c). 

In a release medium with pH 5.5, about 80.6% of TFC was released from PL/ACC-TFC NPs, but only 40% of TFC was released in a release medium with pH 7.4, indicating that TFC could be successfully released from PL/ACC-TFC NPs in a simulated acidic environment (Figure 2e). The pH of the follicular pore decreased from 6.5 to 4.5 with the deepening of the follicles [17]. PL/ACC-TFC NPs triggered more intense drug release in an acidic condition (pH 5.5) than that in a neutral condition (pH 7.4). At the same time, TFC was stable between pH 2.0 and 9.0 with almost no degradation [18]. So, PL/ACC-TFC NPs could release stable TFC from the depot in the form of acid-sensitive trigger-release, diffuse to the hair follicle bulb, and then be absorbed by the follicular epithelial cells and immune cells, which would reduce the frequency of application and effectively enhance the therapeutic effect. 

### 2.2. Determination of Passive Dermal and Follicular Penetration

To improve the therapeutic effect of alopecia areata, it was essential to deliver adequate TFC into the hair follicles [19]. Therefore, we evaluated whether the lipid nanocarriers could transfer more TFC into hair follicles. As shown in Figure 2f,h, the cumulative permeation amount per unit area and steady-state transdermal rate of PL/ACC-TFC NPs were significantly higher than those of the TFC solution group. In addition, PL/ACC-TFC NPs with a particle size of ~280 nm had a higher dermal and follicular permeability and steady-state transdermal rate (Figure 2f,g). The amount of TFC accumulated in the skin of mice in the PL/ACC-TFC group, with a particle size of ~280 nm (12.11 ± 1.68 μg/cm^2^), was 3.15 times higher than that of the TFC solution group (3.84 ± 0.01 μg/cm^2^). The results showed that the active molecule in the TFC solution group mainly gathered outside the skin, and the TFC loaded in PL/ACC NPs could penetrate and accumulate more TFC into the skin. Compared with the TFC solution, PL/ACC-TFC NPs could increase the penetration of TFC into the HF isthmus and establish a drug depot in the HF isthmus, resulting in more drug retention and accumulation in the HF isthmus. 

### 2.3. Dermal Penetration of PL/ACC NPs In Vitro

The preferential accumulation of nanoparticles in hair follicles has been reported in the literature [20]. To obtain visual images of the preferential delivery of active molecules using PL/ACC NPs to hair follicles, porcine ear skin was selected for the transdermal absorption experiment [21]. Rhodamine B (R&B) was loaded into PL/ACC NPs as a fluorescent probe to observe the permeation process of active molecules [22]. The follicular penetration of R&B from the different formulations was monitored via fluorescent images, which analyzed the penetration depth of the nanocarriers into the hair follicles (Figure 3). Fluorescent images of the frozen skin section demonstrated that the accumulation of the red fluorescence of R&B in PL/ACC NPs was significantly higher than rhodamine B solution in the follicular orifice (Figure 3a,b). The mean penetration depth of PL/ACC NPs loaded with R&B into the hair follicles was 700 ± 33 μm, which is equivalent to 65% of regular hair follicle length [23]. The depth of about 300 μm of the hair follicle area represented the lower part of the infundibulum of the hair follicle, where the stratum corneum was weak and easy to penetrate drugs [24]. R&B loaded in PL/ACC NPs showed the strongest fluorescent in this region (Figure 3a), indicating that the ability of PL/ACC NPs to target the infundibulum of the hair follicle opened up the opportunity for effective drug delivery to hair follicles. 

### 2.4. Viability of Cells and Cellular Uptake

Normal mouse fibroblasts (L929) and human hair papillae cells (HDPCs) were used to assess the biocompatibility of PL/ACC NPs. As shown in Figure 4a, the viability of cells was higher than 100%, indicating that PL/ACC NPs as drug carriers have good biocompatibility. More interestingly, when the concentration of PL/ACC exceeded 25 μg/mL, and HDPCs were stimulated and proliferated to about 130%, which was due to the good affinity of phospholipid material for HDPCs, enabling easy absorption by HDPCs [25]. When the TFC concentration was 0.05–5.0 μg/mL, the cell viability of HDPCs was higher than 90%, indicating that TFC was safe and non-toxic for HDPCs (Figure 4b).

HDPCs pretreated with IFN-γ, as an in vitro alopecia areata model, produced a microenvironment similar to alopecia areata [26]. The IFN-γ-induced collapse of hair follicle immune privilege (HF-IP) was used to evaluate the effect of PL/ACC-TFC and TFC solutions on HF-IP. As shown in Figure 4c, IFN-γ (100 ng/mL) significantly down-regulated the cell viability of HDPCs to below 60%, and when different concentrations of PL/ACC-TFC NPs or TFC solution were added, cell viability was restored. The results showed that PL/ACC-TFC NPs and TFC solutions might effectively reverse the IFN-γ-induced collapse of HF-IP. When the concentration of TFC in PL/ACC-TFC or TFC solution was 5 μg/mL, the cell viability of HDPCs pretreated by IFN-γ significantly increased due to TFC stimulating HDPC proliferation. In the cell uptake experiments, the fluorescence intensity of HDPCs co-incubated with PL/ACC-R&B NPs in pH 5.5 medium was significantly higher than that in pH 7.4 medium (Figure 4d,e), which confirmed that pH sensitivity PL/ACC NPs could trigger drug release under acidic conditions.

### 2.5. Hair Follicle Immune Privilege (HF-IP) Restoration/Protection Assay

During the formation of alopecia areata, the IFN-γ-induced collapse of follicular immune privilege resulted in the expression of ectopic MHC class I and II molecules in hair follicular epithelial cells [3]. TFC reduced the secretion of ectopic MHC molecules. Therefore, detecting MHC class I molecules in hair follicles could evaluate the action of TFC. As expected, the hair follicle organ treated with IFN-γ significantly up-regulated the expression of heterotopic MHC class I molecules compared to the control group (Figure 5a,c), indicating that IFN-γ led to follicular immune privilege collapse [27]. In the protection experiments, PL/ACC-TFC NPs or TFC solution was first added to the hair follicle organs before pretreatment via IFN-γ, which significantly reduced the overexpression of heterotopic MHC class I molecules in hair follicle organs (Figure 5c,d). However, PL/ACC-TFC and TFC solutions were ineffective in reducing the expression of ectopic MHC class I molecules in IFN-γ pretreated hair follicle organs in the restoration experiment (Figure 5a,b), suggesting that the protective effect of TFC on hair follicles might be greater than the repair effect. So, PL/ACC NPs with good follicular targeting and skin permeability could increase the concentration of TFC in the hair follicle and slow down or even prevent the progression of AA.

### 2.6. PL/ACC-TFC NPs Controls Alopecia Areata

When the hair follicles in the anagen phase encountered chemotherapy drugs, it led to the miniaturization of hair follicles, accompanied by apoptosis-related damage. The hair follicles then entered the dystrophic catagen stage, leading to broken hair shafts [28]. Therefore, inhibiting cell apoptosis and promoting hair growth were essential for CIA treatment. Genome-wide association study (GWAS) and animal/human studies proved that JAK inhibitors loaded with TFC could antagonize the inflammatory factor IFN-γ in hair follicles, down-regulate the expression of MHC Ⅰ and II molecules, and reduce the number of NKG2D+CD8+ cytotoxic T cells to inhibit apoptosis of hair follicles [29]. It was reported that the inflammatory factor IFN-γ, as one of the by-products of apoptosis-related damage, increased in dystrophic catagen hair follicles in CYP-induced AA [30]. Herein, a CYP-induced alopecia areata mouse model was used to evaluate the control effect of PL/ACC-TFC NPs on alopecia areata [31]. On the 12th day after shaving, the hairs in the dorsal hair loss area of C57BL6 mice injected with CYP significantly fell off, which proved the success of the chemotherapy-induced AA mouse model. Figure 6a shows the treatment schedule for the mice AA model. The changes of hair on the back of the mice during administration were documented weekly, and the results showed that the growth of darkened hairs in the PL/ACC-TFC NPs group was significantly faster than that in the model group and TFC solution groups (Figure 6b). The black hair in the hair loss areas was almost completely covered in the PL/ACC-TFC group, similar to the normal group, with a score of about 5, showing that PL/ACC-TFC almost completely alleviated CYP-induced AA. The TFC solution group covered about 70% of black hair, with a score of about 3. In comparison, the model group only covered about 20% of black hair (Figure 6c). The results showed that both PL/ACC-TFC NPs and TFC solution significantly promoted hair growth, but PL/ACC-TFC NPs had a better effect on alleviating CYP-induced AA than the TFC solution group. 

In addition, the morphological changes of hair follicles stained by HE staining on the longitudinal section of the skin in each group (Figure 6d) showed that the melanin granules of the normal group of mice were apparent and located under the skin, and the hair follicles were in the anagen phase. On the contrary, the melanin granules in the model group were destroyed, and the hairs grown were gray, indicating that the hair follicles were in the dystrophic anagen phase. The TFC solution group had only partially intact melanin granules, indicating that the TFC solution also somewhat alleviated CYP-induced AA. The results suggested that PL/ACC NPs effectively delivered TFC into hair follicles (hair bulge), thereby enhancing hair growth effects.

The previous studies proved that the pathogenesis of AA was associated with the accumulation of NKG2D+CD8+ cytotoxic T cells, resulting in the collapse of hair follicle immunity and hair loss [32]. The expression of MHC class II molecules in the model group was significantly higher than that in the other groups (Figure 6e), which also consisted of the requirements of the mouse model of AA. As shown in Figure 6e, the expression of MHC-II molecules in the PL/ACC-TFC NPs group was significantly lower than that in the TFC solution group. NKG2D+CD8+ cytotoxic T cells were detected in the dermis of mice in the model group through flow cytometry, with a significantly higher number than other groups (Figure 6f and Appendix A). These results further prove the success of the mouse CYP-induced AA model. After topical treatment, compared with the model group, the expression of MHC-II molecules in the dermal papilla cells in the PL/ACC-TFC group decreased, and the number of NKG2D+CD8+ cytotoxic T cells around the dermal papilla cells also reduced. PL/ACC-TFC NP had a better therapeutic effect on CYP-induced AA.

## 3. Materials and Methods

### 3.1. Materials

Ammonium carbonate (NH_4_HCO_3_), anhydrous calcium chloride (CaCl_2_), sodium dodecyl sulfate (SDS), chloroform, ethanol, and glyceryl monostearate were purchased from Sinopharm Chemical Reagent Co, Ltd. (Shanghai, China). Tofacitinib citrate (TFC) was obtained from Wuhan Xiangheshunda Biotechnology Co., Ltd. (Wuhan, China). Soybean lecithin was purchased from Shanghai McLean Biochemical Technology Co., Ltd. (Shanghai, China). Rhodamine B was obtained from Sigma-Aldrich (St. Louis, MO, USA). Cyclophosphamide Injection was purchased from Baxter Oncology GmbH (Kantstr. Halle, Germany). The reagents used for cell and animal experimental evaluation were shown in the Appendix A. Other conventional reagents, including AR, were purchased from Sinopharm Chemical Reagent Co., Ltd. (Shanghai, China). 

### 3.2. Cell Line

Mouse fibroblast cells L929 were purchased from CCTCC (Wuhan, China). Human dermal papilla cells (HDPCs) and Human hair follicle organs (HHFs) were provided by Dr. Hanluo Li of Hubei University of Technology. DMEM complete medium containing 10 % (*v*/*v*) FBS and 1% (*v*/*v*) P/S were used to culture L929 cells in a cell incubator at 37 °C in a 5% CO_2_ atmosphere.

### 3.3. Preparation of Lipid-Coated Calcium Carbonate Nanocarrier (PL/ACC-TFC)

ACC NPs with three different particle sizes were obtained through the gas diffusion method based on different formulations (Appendix A), and PL/ACC-TFC NPs with different particle sizes were prepared using three templates of ACC NPs with different particle sizes. A certain amount of CaCl_2_ was dissolved in ethanol (50 mL) containing different volumes of water, then the mixture was transferred to a small beaker packed with a porous membrane. The conical flask containing NH_4_HCO_3_ powder (2.5 g) and the beaker containing CaCl_2_ solution were placed in a sealed beaker at 40 °C for 24 h and produced ACC NPs through the gas diffusion method. ACC NPs with three different particle sizes were separated by centrifugation (10,000 rpm/15 min), dispersed into anhydrous ethanol again, and collected by drying in a vacuum oven at 60 °C for 2 h. Under stirring, three samples of tofacitinib citrate (TFC, 10 mg) were, respectively, dissolved in anhydrous ethanol (10 mL) at 60 °C to obtain the TFC solution. ACC NPs with three different particle sizes (40 mg) were added to TFC solution under ultrasonic using Ultrasonic Cleaners (SB-5200DTD, SCIENTZ, China) for 10 min, respectively. Then, the mixture was stirred at 40 °C for 2 h to obtain calcium carbonate nanoparticles loaded with tofacitinib citrate (ACC-TFC NPs). Subsequently, soy lecithin (40 mg) and triglyceride monostearate (20 mg) were dispersed in 10 mL trichloromethane to obtain a lipid solution, which was, respectively, added to ACC-TFC NPs with three different particle sizes using a constant pressure funnel and stirred at 40 °C for 2 h. After that, the mixture was evaporated at 40 °C (Rotavapor R-3, Buchi, Switzerland) for 30 min to completely remove the organic solvent and form a drying film on the bottle wall. The film was dissolved in 1.5% SDS (10 mL, *m*/*v*) to obtain a lipid-coated calcium carbonate nanocarrier (PL/ACC-TFC NPs) with three different particle sizes. 

Rhodamine B was chosen as the fluorescent probe to replace TFC. Lipid nanoparticles loaded with rhodamine B (denoted as PL/ACC-R&B) were prepared in a similar manner to PL/ACC-TFC [22]. In addition, rhodamine B (10 mg) was dissolved in a solution containing 1.5% SDS (10 mL) to obtain rhodamine B solution (R&B solution).

### 3.4. Characterization of PL/ACC-TFC NPs

Particle size, polydispersity index (PDI), zeta potential of ACC NPs, ACC-TFC, and PL/ACC-TFC were analyzed using Zetasizer Nano ZS90 (Malvern Instruments, Worcestershire, UK), respectively. In order to calculate the drug loading (DL%) and encapsulation efficiency (EE%) of PL/ACC-TFC NPs, the concentration of TFC was detected via High-Performance Liquid Chromatography (HPLC) analysis (Dionex U3000 system, Dionex Technologies Inc., Sunnyvale, CA, USA, Appendix A). TFC in PL/ACC-TFC NPs was detected under the guidance of TFC standard curve (Appendix A). PL/ACC-TFC was morphologically characterized using a transmission electron microscope (TEM) at an accelerated voltage of 200 kV with different magnifications (Tecnai G220S-TWIN, Eindhoven, The Netherlands).

The stability of PL/ACC-TFC NPs at 4 °C at a series of time points (day 1, 3, 5, 7, 15, and 30 days) was monitored through the change in particle size and polydispersity index (PDI). Drug release was evaluated under different pH conditions (pH 5.5/6.5/7.4). PL/ACC-TFC NPs (1 mL) were added into dialysis bags (MWCO: 3500 Da), which were immersed in 150 mL PBS (0.1 M, pH5.5/6.5/7.4) and stirred continuously at 100 rpm at 37 °C. The release medium (2 mL) was removed at selected time intervals and replenished with the same volume of fresh medium. The concentration of TFC in the release medium was measured using a UV-vis spectrometer (UV-1800, Shimadzu, Japan) at 287 nm.

### 3.5. Evaluation of PL/ACC-TFC NPs Delivery In Vitro

#### 3.5.1. Ex Vivo Model of Dermal and Follicular Penetration

Fresh porcine ears were obtained from the local slaughterhouse and used within a few hours after slaughter. Prior to treatment, pig ears were rinsed in cold water and dried with a tissue. Three pieces of 2 × 2 cm^2^ intact skin (marked as areas 1, 2, and 3) without visible wounds were selected, and the hair in the regions was cut to approximately 1~3 mm with scissors. Three samples (100 μL/sample), including PBS, R&B solution, and PL/ACC-R&B NPs, were applied to areas 1, 2, and 3 of the porcine ears, respectively. After penetration at 32 °C for 6 h, the excess samples on the skin were carefully washed off, the skin tissue was taken from areas 1, 2, and 3 of the porcine ears (using a scalpel), scraped off the subcutaneous fat, and rapidly frozen in liquid nitrogen. After freezing, the skin tissues were embedded on circular specimen discs with a diameter of 2.2 cm using an OCT embedding agent, and skin sections (20 μm/section) were prepared using a cryo-microtome (CM1860 UV, Leica, London, UK). The fluorescence distribution in the cryo-sections was observed using an inverted fluorescence microscope (IX73, Olympus, Shinjuku, Japan).

#### 3.5.2. Investigation of TFC Delivery to Skin In Vitro

In vitro penetration experiment was performed using a vertical-type Franz diffusion cell (TK-12A; Huanghai Pharmaceutical Testing Instrument Co., Ltd., Shanghai, China). The dorsal skin of mice (2.54 cm^2^), as the diffusion skin, was fixed between the supply and receiving pools with the epidermal layer of the skin facing upward. The receptor compartment was filled with 7.5 mL of PBS (0.1 M, pH = 6.8) and maintained at 32 °C under constant magnetic stirring at 800 rpm. PL/ACC-TFC NPs (1 mL) with different particle sizes in Appendix A and 1% TFC solution (1 mL, *w*/*v*) were applied to the skin topically. The media (2 mL every time) in the acceptor chamber was taken out and supplemented with the same volume of fresh media during time intervals of 0.5 h, 1 h, 2 h, 4 h, 8 h, 12 h, and 24 h. The content of TFC in the release medium was measured by HPLC, and the cumulative permeation amount per unit area and steady-state transdermal rate per unit area were calculated, as shown in Appendix A [33].

After 24 h of transdermal experiments, TFC content per unit area of skin in different treatment groups was monitored. First, the skin was removed from the diffusion cell and rinsed with deionized water before being dried and weighed. The skin sample and deionized water (2 mL) were added to a centrifuge tube and sonicated 30 times using an ultrasonic cell disruptor (DR500 Std, DISRAD, St. Diego, CA, USA) probe, then all samples were highly sheared at 10,000 rpm for 10 min. Dimethyl sulfoxide (1 mL) and methanol (2 mL) were added to the centrifuge tube to extract TFC and heated at 60 °C for 30 min. The supernatant was extracted by centrifugation at 10,000 rpm for 10 min, and the TFC content was determined by HPLC (Appendix A).

### 3.6. Cell Viability and Cellular Uptake Test

To explore the biocompatibility of the PL/ACC NPs, TFC solution, and PL/ACC-TFC NPs in vitro, the cell viability was measured via MTT assay using L929 cells and HDPCs. Briefly, the series concentration of PL/ACC NPs, TFC solution, and PL/ACC-TFC NPs were added to the 96-well plate containing 1 × 10^4^ L929 or HDP cells/well and incubated for 24 h. Then, MTT (5 mg/mL, 20 μL) was added to each well and cultured at 37 °C for 4 h, and the methylzane crystals were dissolved by the addition of 150 μL DMSO. The optical density at 490 nm was measured using a microplate reader (BioteK, Epoch, Winooski, VT, USA), and the percentage cell viability was calculated via the following formula: Percentage cell viability = (absorbance of experimental well—absorbance of blank)/(absorbance of untreated control well—absorbance of blank) × 100%.

To investigate whether TFC could increase the viability of HDPCs, different concentrations of TFC solution or PL/ACC-TFC NPs and IFN-γ (100 ng/mL) were added to the 96-well plate, and the cell viability of HDPCs was evaluated in the same way as above. To further evaluate the pH sensitivity of PL/ACC-TFC NPs, Rhodamine B was used as a substitute for TFC in the cellular uptake test. Firstly, different pH of cell medium (pH 5.5 and 7.4) were prepared with lactic acid [15]. PBS, R&B solution, PL/ACC-R&B NPs (pH = 5.5), and PL/ACC-R&B NPs (pH = 7.4) with a concentration of R&B 50 μg/mL were added to the 24-well plate containing 1 × 10^5^ HDPCs/well, respectively, and the uptake of each group of cells was observed with an inverted fluorescence microscope at 1 h, 3 h, and 6 h, respectively.

### 3.7. Hair Follicle Immune Privilege (HF-IP) Restoration/Protection Assay

#### 3.7.1. Hair Follicle (HF) Organ Culture Assay

Intact growth type IV hair follicles (HFs) of a normal 28-year-old male healthy human scalp were cultured in 24-well culture plates using William’s E medium, with 3 hair follicles per well. The culture medium was 500 μL William’s E medium containing bovine insulin (10 μg/mL), hydrocortisone (10 ng/mL), streptomycin (100 μg/mL), and penicillin (100 U/mL). In accordance with processes described in the literature [34], HFs were pretreated with IFN-γ100 (ng/mL) for 4 days to induce immune privilege collapse as the repair experiment model, and blank medium, TFC solution, and PL/AC-TFC NPs were added to incubate for 2 days, respectively. On the other hand, HFs were firstly treated with a blank medium, TFC solution, or PL/ACC-TFC NPs for 2 days, followed by the addition of IFN-γ (ng/mL) to incubate for 4 days as the protection experiment model.

#### 3.7.2. Immunohistochemistry Staining

The MHC class I molecules in the hair follicles organ were detected via immunohistochemical staining using the method in Appendix A. The frozen section of hair follicle organ with 8 μm thickness was subjected to immunohistochemical staining, and the hair follicles were visualized using a BZ-X810 All-in-One fluorescence microscope and analyzed with Image J software (Image J 1.4.3).

### 3.8. Application for Cyclophosphamide (CYP)-Induced Alopecia Areata

#### 3.8.1. Hair Regrowth Assays

Female C57BL/6 mice (6-week-old) were provided by the Laboratory Animal Center, Hubei Academy of Preventive Medicine (Wuhan, China). The care and use of mice followed the guidelines of the IACUC committees of the Hubei University of Technology. CYP-induced alopecia areata model mice were performed as described previously [35]. The mice were anesthetized via intraperitoneal injection of 3% phenobarbital solution. The dorsal skin hair of 6 cm^2^ area (length, 3 cm; width, 2 cm) was removed carefully using a shaver, then a small amount of depilatory cream was applied to induce anagen hairs. On the 7th day after epilation, the dorsal skin after hair removal started to darken, indicating that the hair follicles were in the anagen phase. A single intraperitoneal injection of freshly dissolved CYP in saline (150 mg/kg body weight) was administered to the mice, except for the normal mice group. Five days after cyclophosphamide injection, the mice started to lose hair on the head and back, which indicated that the CYP-induced alopecia areata model had been successfully prepared. The treatment schedule on the mice AA model was shown in Figure 6a. The CYP-induced alopecia areata model mice were randomly divided into three groups as follows: Control model, TFC solution, and PL/ACC-TFC NPs group. An amount of different formulation was topically administered to the depilated region on days 1, 7, 14, and 21. The normal and Control group received normal saline during the experiment, while the TFC solution and PL/ACC-TFC NP group were administered with 1% TFC solution and 1% PL/ACC-TFC (100 μL/body), respectively. The hair changes on the back of mice were photographed daily for 21 days of continuous administration. The hair regrowth was scored in terms of the thickness and color of hair growth for each group of mice [36,37]. The scoring criteria were as follows: grade 0: no regeneration of new black hairs at all; grade 1: 0–20% of the total hair loss area of new black hairs; grade 2: 20–40% of new black hairs regenerated; grade 3: 40–60% of new black hairs regenerated; grade 4: 60–80% of new black hairs regenerated; grade 5: 80–100% of new black hairs regenerated. The density of black hair was measured using image analysis software (ImageJ, Version 1.50i).

#### 3.8.2. Histological Staining of Skin Samples

On the 21st day, the skin on the back of the mice was taken after neck-breaking and execution of the mice. The skin was fixed overnight using 4% paraformaldehyde (pH 7.4), dehydrated, paraffin-embedded, and sectioned at 8 μm using a Leica paraffin microtome (JY-QPA, Wuhan, China). Sections were dewaxed with xylene, rehydrated with decreasing concentrations of 100–50% ethanol, and then washed with distilled water. Skin sections were stained with hematoxylin and eosin, washed with tap water, and finally sealed with neutral gum. The MHC class II molecules in the hair follicles of the dorsal skin of experimental mice were detected via immunohistochemical staining using the method in Appendix A. The frozen section of dorsal skin with 8 μm thickness was subjected to immunohistochemical staining. The hair follicles were visualized using a BZ-X810 All-in-One fluorescence microscope and analyzed with Image J software.

### 3.9. Flow Cytometry

To prepare skin single-cell suspensions, the dorsal skin of the mice in different treatment groups was digested with trypsin at 37 °C for 20 min, the skin dermis was scraped off using pliers and a scalpel, then the skin tissue was cut with scissors and incubated with DMEM medium containing collagenase type IV (2 mg/mL) and DNase (1 μg/mL) for 20 min at 37 °C, then sheared with 10,000 rpm for 5 min. The digested skin was passed over a 70-μm cell strainer, followed by the removal of erythrocytes from splenocytes with ACK lysis buffer, incubated on ice for 15 min, and collected by centrifugation and resuspension. Skin single cell suspension was incubated with 1.5% BSA in PBS on ice for 1 h, APC-CD8a antibody and NKG2D antibody were added according to the manufacturer’s requirements and incubated for 30 min at 4 °C. NKG2D+CD8+ T cells in the samples were analyzed using flow cytometry software BD AccuriC6 (BD Accuri C6-Biosciences, San Francisco, CA, USA).

### 3.10. Statistical Analysis

All quantitative results were expressed as mean ± standard deviation (SD). The data were statistically analyzed using the GraphPad Prism 8.0 program. The significance of difference between the means of three independent experiments was determined by ANOVA—ns means no significant differences, while * *p* < 0.05, ** *p* < 0.01, *** *p* < 0.001 and **** *p* < 0.001 were considered significant.

## 4. Conclusions

In conclusion, we successfully prepared PL/ACC-TFC NPs with good biocompatibility and stability. These calcium carbonate-based lipid nanoparticles could enhance the dermal permeability and target the hair follicle. In a pig ear model, PL/ACC NPs maximized the targeted delivery of active molecules into hair follicles in vitro, which increased the concentration of drugs in the target site and decreased the adverse effects of TFC. In addition, PL/ACC NPs provided an effective method to target hair follicles, delivered TFC in CYP-induced Alopecia areata mouse model, antagonized the inflammatory factor IFN-γ on inducing the collapse of immune immunity in hair follicles, and down-regulated the expression of MHC II molecules and the number of NKG2D+CD8+ cytotoxic T cells to inhibit apoptosis of hair follicles. These results reveal that PL/ACC-TFC NPs could induce hair growth and have the potential to be applied in the prevention and mitigation of CYP-induced Alopecia areata.

## Figures and Tables

**Figure 1 ijms-24-08427-f001:**
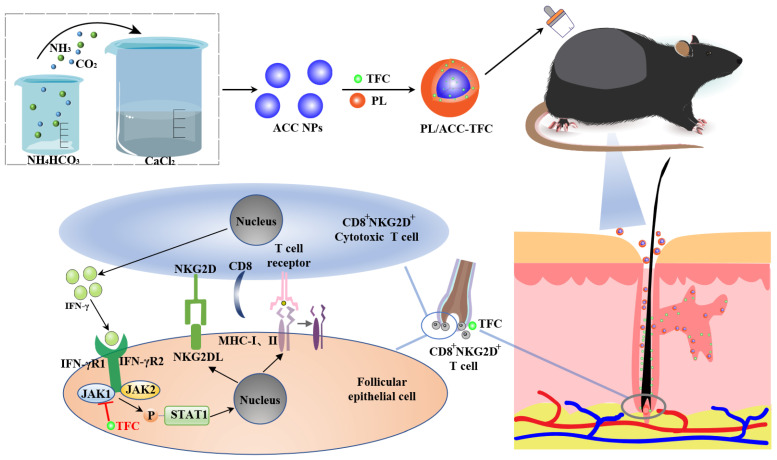
Schematic illustration of PL/ACC-TFC NPs targeted delivery to hair follicles for the treatment of CYP-induced Alopecia areata.

**Figure 2 ijms-24-08427-f002:**
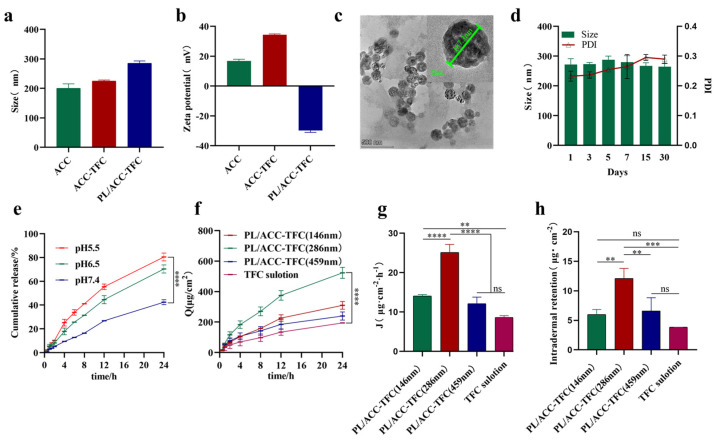
Characterization of PL/ACC-TFC NPs. (**a**) The particle size of ACC NPs, ACC-TFC NPs, PL/ACC-TFC NPs. (**b**) Zeta potential of nanocarriers. (**c**) TEM of PL/ACC-TFC NPs. (**d**) Stability of PL/ACC-TFC over 30 days. (**e**) In vitro drug release of PL/ACC-TFC at different pH. (**f**) The cumulative permeation amount per unit area of PL/ACC-TFC NPs with different sizes. (**g**) Steady-state transdermal rate per unit area in different groups. (**h**) TFC content in different groups per unit area of skin. Data are expressed as mean ± SD, *n* = 3. (ns, not significant, ** *p* < 0.01; *** *p* < 0.001; **** *p* < 0.0001).

**Figure 3 ijms-24-08427-f003:**
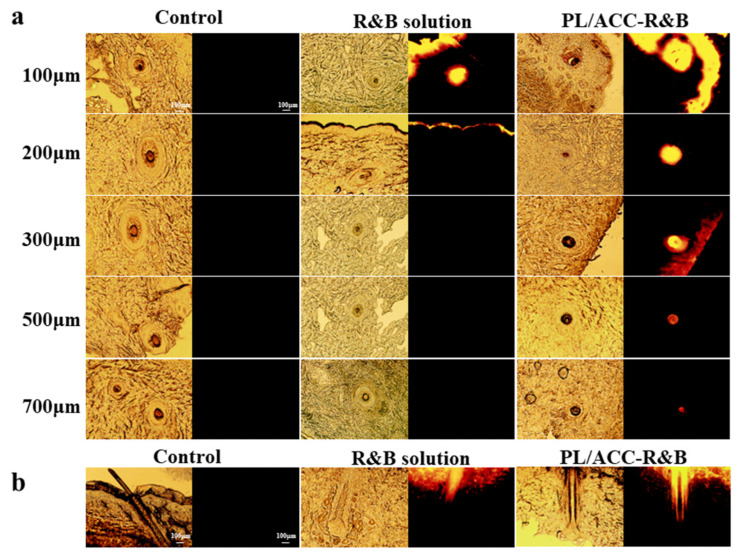
Visualization of R&B formulation into the hair follicles of porcine. (**a**) Fluorescence distribution of R&B solution or PL/ACC NPs in the cross-section. (**b**) Fluorescence distribution of R&B solution or PL/ACC NPs in the longitudinal section. (Left: bright field diagram; Right: fluorescence diagram; Scale bars represent 100 μm).

**Figure 4 ijms-24-08427-f004:**
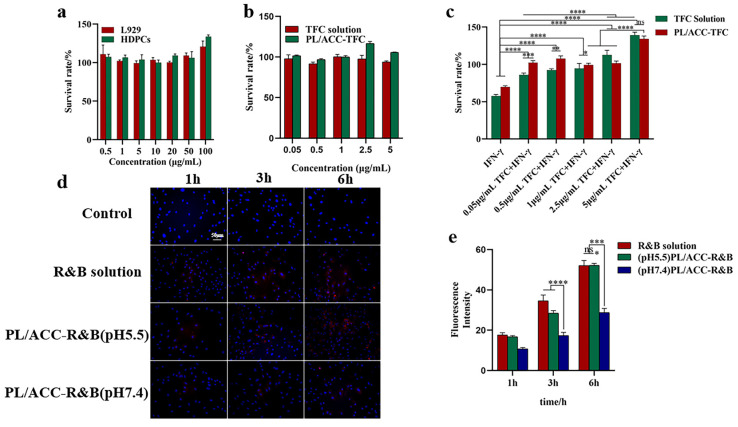
Biocompatibility and cellular uptake in vitro. (**a**) Biocompatibility evaluation of PL/ACC NPs on L929 and HDPCs cells. (**b**) Biosafety of TFC solution and PL/ACC-TFC on HDPCs at 24 h. (**c**) Effect of PL/ACC-TFC and TFC solutions on IFN-γ-induced collapse of hair follicle immune privilege (HF-IP). (**d**) Cell uptake of R&B by HDPCs at different pH. (**e**) Histogram for quantification of fluorescence intensity using Image J software. Data are expressed as mean ± SD, *n* = 3. Scale bars represent 50 μm. (* *p* < 0.05; ** *p* < 0.01; *** *p* < 0.001 and **** *p* < 0.0001; ns, not significant).

**Figure 5 ijms-24-08427-f005:**
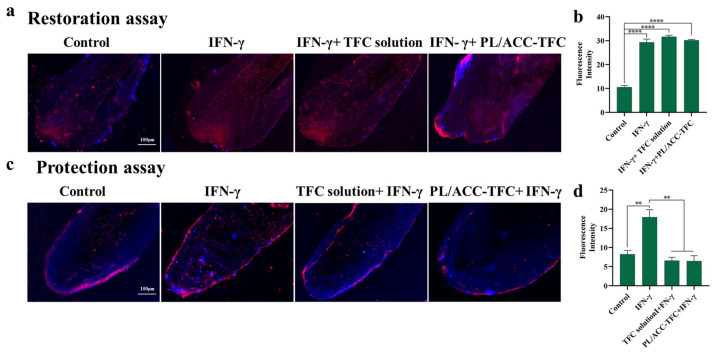
Protection and repair evaluation of hair follicle organ. (**a**) Restoration experiments. (**b**) Quantification of restoration experiments. (**c**) Protection experiments. (**d**) Quantification of protection experiments. The scale bar represents 100 μm. (** *p* < 0.01 and **** *p* < 0.0001).

**Figure 6 ijms-24-08427-f006:**
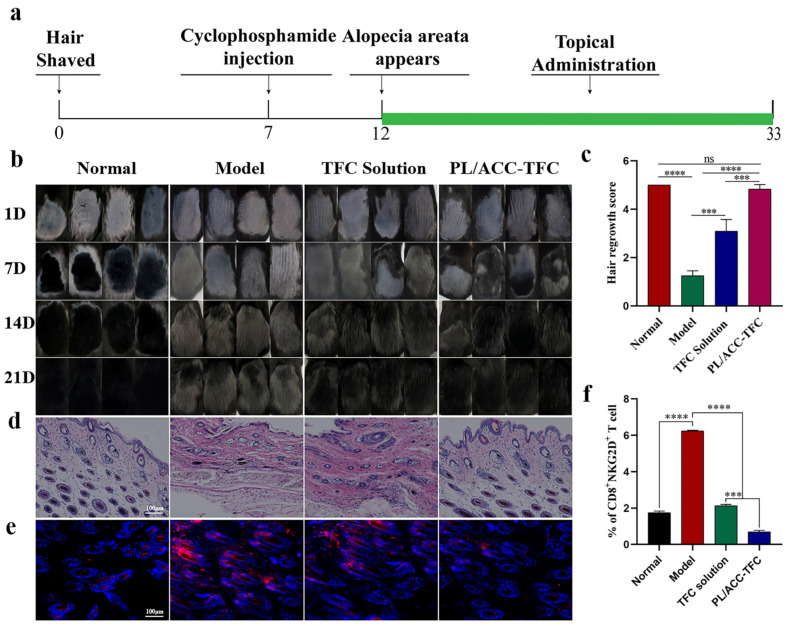
The topical application of PL/ACC-TFC NPs control CYP-induced alopecia areata. (**a**) The treatment schedule for the CYP-induced alopecia areata mouse model. (**b**) Graphs of skin changes in model mice after drug administration. (**c**) Skin hair regeneration scores on the 21st day. (**d**) Longitudinal sections of skin in different treatment groups. (**e**) Immunohistochemical detection of MHC class II in frozen sections. (**f**) Flow cytometry detection of NKG2D+CD8+ T cell levels in different treatment groups. Data are expressed as mean ± SD, *n* = 3. Scale bars represent 100 μm. (*** *p* < 0.001 and **** *p* < 0.0001; ns, not significant).

## Data Availability

No new data were created.

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
