# Peer review of "Selective Delivery of Tofacitinib Citrate to Hair Follicles Using Lipid-Coated Calcium Carbonate Nanocarrier Controls Chemotherapy-Induced Alopecia Areata"

_ijms, 2023, doi:10.3390/ijms24098427_

Round 1

Reviewer 1 Report

The paper by Zhu et al. describes the use of calcium, carbonate nanoparticles for the delivery of Tofacitinib to alleviate alopecia areata induced by chemotherapy treatment. The paper deals with an important topic and it deserves attention considering the severe distress caused by hair loss considering the effect on patients’ compliance to chemotherapy. Some minor changes are necessary in my opinion before publication.

First of all, the paper needs a proof-reading by a native English speaker in order to increase the quality of manuscript.

1.       in the Abstract: CYP acronym should be clarified

2.       lines 106-110; 118-119; 158-160;215 -216; 296-299 for instance should be rephrased.  Lines 42-42: “alleviate” is repeated twice in close proximity

3.       in the Introduction section: please expand the effect of the release of MHC II in the development of alopecia.

4.       Figure 1: It’s not clear in my opinion the effect and the targets of TFC in JAK/STAT pathway

5.       paragraph 2.1: please clarify the conditions under which the authors tested the stability during 30-day-storage.

6.       I encourage the authors to test (or to introduce it, if the authors have already gathered this information) the TFC stability under the same different pH conditions (pH 5.5, 6.5, 7.4) used for evaluating the release from PL/ACC NPs. This aspect is important to establish the real pH-sensitivity of the NPs in the sustained release of the active ingredient.

7.       paragraph 2.5: please explain in a more detailed way the meaning of Figures 5a-5c

Author Response

We are truly grateful for your valuable comments and hard work. Based on the comments and suggestions of reviewer and editor, we have made carefully modifications to the original manuscript. The comments provided will help in improving our manuscript. Below, we have summarized a point-to-point response to the reviewers’ comments. All modifications were red highlighted in the revised manuscript.

We appreciate the critical reviews of the manuscript and hope the revised version is improved to a significant degree for publication

  1. in the Abstract: CYP acronym should be clarified

Response: Thank you for your suggestion. The abbreviation for CYP comes from cyclophosphamide, which has been added to the abstract and marked in red.

  1. lines106-110; 118-119; 158-160; 215-216; 296-299 for instance should be rephrased. Lines 42-42: “alleviate” is repeated twice in close proximity

Response: Thank you for pointing out these problems. Based on your suggestion, we have rephrased these sentences and marked them in red.

  1. in the Introduction section: please expand the effect of the release of MHC II in the development of alopecia.

Response: Thank you for your suggestion. During the formation of CIA, the inflammatory factor IFN-γ secreted by CD8+NKG2D+ T cells increased due to chemotherapy stimulation. IFN-γ could bind to the corresponding receptors of hair follicle epithelial cells, resulting in JAK1/2-mediated STAT1 phosphorylation, thereby causing the hair follicle epithelial cell nucleus to secrete more ectopic MHC class I and II molecules. These molecules would recruit more CD8+NKG2D+ T cells to attack hair follicles and lead to hair loss [1]. As suggested by the reviewers, we have added this description in the introduction section of the manuscript and marked them in red.

  1. Figure 1: It’s not clear in my opinion the effect and the targets of TFC in JAK/STAT pathway.

Response: Thank you for your suggestion. During the formation of alopecia areata, the inflammatory factor IFN-γ secreted by CD8+NKG2D+ T cells would result in follicular immune privilege and JAK1/2-mediated STAT1 phosphorylation and secrete more ectopic MHC class I and II molecules on hair follicular epithelial cells. TFC, as small-molecule JAK1/3 inhibitors, could interfere with this feedback loop and upregulate the expression of ectopic MHC class I and II molecules, reducing the recruitment of CD8+NKG2D+T cells and preventing the progression of hair loss [2].

  1. paragraph 2.1: please clarify the conditions under which the authors tested the stability during 30-day-storage.

Response: We evaluated the stability of PL/ACC-TFC stored at 4 ° C for 30 days and added storage conditions, which were marked in red in the revised manuscript.

  1. I encourage the authors to test (or to introduce it, if the authors have already gathered this information) the TFC stability under the same different pH conditions (pH 5.5, 6.5, 7.4) used for evaluating the release from PL/ACC NPs. This aspect is important to establish the real pH-sensitivity of the NPs in the sustained release of the active ingredient.

Response: We thank the reviewer for pointing out the issue of whether the stability of TFC under acidic conditions will affect its activity? Younis et al. [3] proved TFC was stable between pH 2.0 and 9.0 with almost no degradation, and remained stable for over 700 days at pH 7.

  1. paragraph 2.5: please explain in a more detailed way the meaning of Figures 5a-5c

Response: Thank you for your suggestions. In the protection experiments, PL/ACC-TFC NPs or TFC solution was first added to the hair follicle organs before IFN-γ pretreated, which could significantly reduce the overexpression of heterotopic MHC class I in hair follicle organs (Figure 5c and 5d). However, PL/ACC-TFC and TFC solutions were ineffective in reducing the expression of ectopic MHC class I in IFN-γ pretreated hair follicle organs in the restoration experiment (Figure 5a and 5b), suggesting that the protective effect of TFC on hair follicles might be greater than the repair effect. 

Reference

  1. Xing L, Dai Z, Jabbari A, Cerise JE, Higgins CA, Gong W, de Jong A, Harel S, DeStefano GM, Rothman L, et al: Alopecia areata is driven by cytotoxic T lymphocytes and is reversed by JAK inhibition. Nat Med 2014, 20:1043-1049.
  2. Divito SJ, Kupper TS: Inhibiting Janus kinases to treat alopecia areata. Nat Med 2014, 20:989-990.
  3. Younis US, Vallorz E, Addison KJ, Ledford JG, Myrdal PB: Preformulation and Evaluation of Tofacitinib as a Therapeutic Treatment for Asthma. AAPS PharmSciTech 2019, 20:167.

Reviewer 2 Report

Yeneng Guan and co-workers describe a lipid nanosystem that allows efficient delivery of Tofacitinib citrate to hair follicles, and prevents hair loss during cytostatic treatment of cancer. The manuscript is potentially interesting, but perhaps more adequate for a Pharmacy-oriented journal.

Author Response

Reviewer #2:

Yeneng Guan and co-workers describe a lipid nanosystem that allows efficient delivery of Tofacitinib citrate to hair follicles, and prevents hair loss during cytostatic treatment of cancer. The manuscript is potentially interesting, but perhaps more adequate for a Pharmacy-oriented journal.

Response: Thank you for your suggestions. Our manuscript was submitted to the special issue “Future Perspectives in Nanostructured Materials Preparation, Characteristics and Applications” and “Nano Based Drug Delivery Systems: Future and Development” in the International Journal of Molecular Sciences Materials Science section. So, the content is suitable for this special issue.

Reviewer 3 Report

Very interesting paper that deserves to be published.

1)      The sentence in lines 109-110: “NPs with the particle size of about 280 nm had better dermal and follicular permeability 109 than other size nanocarriers (Figure 2f). “Which other mean sizes ?  This sentence needs to be improved.

   2) The  method used for Nanoparticles should be improved. Some details regarding their preparation should be included. For example, although in materials and methods the amounts of phospholipids used are included, experimental conditions for achieving lipid-coated calcium carbonate nanocarrier (PL/ACC-TFC NPs) with different particle sizes are missing.

 3) Lines 393-396: “The  content of TFC in the release medium was measured using a UV-vis spectrometer, and the cumulative permeation amount per unit area and steady-state transdermal rate per  unit area…” However in lines 403-404  you refer that  “TFC was determined by HPLC”. Please clarify.

4) Lines 410-411 “The cell viability was measured using a microplate reader (BioteK, Epoch, USA) after incubation for 24 h.”. Which was the method?

5)      Line 106 – “Some researchers demonstrated that lipid nanoparticles particles with the size of…” Please check the sentence.

6)      Is it possible to improve the quality of figure 4 d) ?

7)   The English should be revised in all manuscript.

Author Response

We are truly grateful for your valuable comments and hard work. Based on the comments and suggestions of reviewer and editor, we have made carefully modifications to the original manuscript. The comments provided will help in improving our manuscript. Below, we have summarized a point-to-point response to the reviewers’ comments. All modifications were red highlighted in the revised manuscript.

We appreciate the critical reviews of the manuscript and hope the revised version is improved to a significant degree for publication.

Reviewer #3:

Very interesting paper that deserves to be published.

  1. The sentence in lines 109-110: “NPs with the particle size of about 280 nm had better dermal and follicular permeability 109 than other size nanocarriers (Figure 2f). “Which other mean sizes? This sentence needs to be improved.

Response: The sentence "other mean sizes" refers to PL/ACC-TFC NPs with particle sizes of about 150 nm and 460 nm. Thank you for the suggestion. We revised it and marked in red.

  1. The method used for Nanoparticles should be improved. Some details regarding their preparation should be included. For example, although in materials and methods the amounts of phospholipids used are included, experimental conditions for achieving lipid-coated calcium carbonate nanocarrier (PL/ACC-TFC NPs) with different particle sizes are missing.

Response: Thank you for your suggestions. PL/ACC-TFC NPs with different particle sizes were prepared using three templates ACC NPs with different particle sizes. Here, ACC NPs with three different particle sizes were prepared through the gas diffusion method based on different formulations (Supporting information Table S2). We added the method in Section 3.3.

  1. Lines 393-396: “The content of TFC in the release medium was measured using a UV-vis spectrometer, and the cumulative permeation amount per unit area and steady-state transdermal rate per unit area…” However, in lines 403-404 you refer that “TFC was determined by HPLC”. Please clarify.

Response: Sorry, an expression error occurred in this sentence. Here, TFC content in vitro drug release experiments was determined using a UV-vis spectrometer. The TFC content in the transdermal experiment was determined by HPLC. The error has been corrected in the revised manuscript and marked in red.

  1. Lines 410-411 “The cell viability was measured using a microplate reader (BioteK, Epoch, USA) after incubation for 24 h.”. Which was the method?

Response: The cell viability was measured using the MTT assay. The experimental proceed have been added in the revised manuscript. The series concentration of PL/ACC NPs, TFC solution and PL/ACC-TFC NPs were added into the 96-well plate containing 1×104 L929 or HDP cells/well. Then, MTT (5 mg/mL, 20 μL) was added to each well and cultured at 37°C for 4 h, the methylzane crystals were dissolved by the addition of 150 μL DMSO. The optical density at 490 nm was measured using a microplate reader (BioteK, Epoch, USA) and the percentage cell viability was calculated via the formula: Percentage cell viability = (ab-sorbance of experimental well - absorbance of blank)/ (absorbance of untreated control well - absorbance of blank) × 100%.

  1. Line 106 – “Some researchers demonstrated that lipid nanoparticles particles with the size of…” Please check the sentence.

Response: Thank you for your suggestion. We revised this sentence in the revised manuscript and marked in red.

  1. Is it possible to improve the quality of figure 4 d)?

Response: Thank you for your suggestion. According to the reviewers' suggestions, we improved the quality of Figure 4d in the revised manuscript.

  1. The English should be revised in all manuscript.

Response: We deeply apologize for any inconvenience caused to your reading of this manuscript. We have made every effort to revise this manuscript, and these changes marked in red would not influence the content and framework of the manuscript. We hope that the revised manuscript meets the requirements. Thanks.